

# The implementation of NEMS GFS Aerosol Component (NGAC) Version 2.0 for global multispecies forecasting at NOAA/NCEP: Part II Evaluation of Aerosol Optical Thickness

Partha S. Bhattacharjee[1], Jun Wang[1], Cheng-Hsuan Lu[2] and Vijay Tallapragada[3]

[1]I.M. Systems Group at NOAA/NWS/NCEP/EMC, College Park, 20740, USA
[2]University of Albany, State University of New York, Albany, 12222, USA
[3]NOAA/NWS/NCEP/EMC, College Park, 20740, USA

*Correspondence to*: Partha S. Bhattacharjee (partha.bhattacharjee@noaa.gov)

**Abstract**

An accurate representation of aerosols in global Numerical Weather Prediction (NWP) models is important to predict major air pollution events and to also understand aerosol effects on short-term weather forecasts. Recently the global aerosol forecast model at NOAA, the NOAA Environmental Modeling System (NEMS) GFS Aerosol Component (NGAC), was upgraded from its dust-only version 1 to include five species of aerosols (black carbon, organic carbon, sulfate, sea-salt and dust). This latest upgrade, now called NGACv2, is an in-line aerosol forecast system providing 3-dimensioanl aerosol mixing ratios along with aerosol optical properties, including aerosol optical thickness (AOT), every 3 hours up to 5 days at global 1°x1° resolution. In this paper, we evaluated nearly one and half years of model AOT at 550nm with available satellite retrievals, multi-model ensembles and surface observations over different aerosol regimes. Evaluation results show that NGACv2 has high correlations and low root mean square errors associated with African dust and also accurately represented the seasonal shift of aerosol plumes from Africa. Also, the model represented South African and Canadian forest fires, dust from Asia and AOT within the US with some degree of success. We have identified model underestimation for some of the aerosol regimes (particularly over Asia) and will investigate this further to improve the model forecast. The addition of a data assimilation capability to NGAC in the near future is expected to improve some of the model biases.

## 1. Introduction

In the past two decades, aerosol distributions, their properties and their impact have been studied using a combination of complex numerical models and space and ground-based monitoring programs. Aerosols play a crucial role in climate and the hydrologic cycle by altering the radiation balance and clouds. Also, large concentrations of aerosol particles near the surface influence ambient air quality and human health (Menon et al., 2002). Natural and anthropogenic aerosols are thought to play an important role in global climate model projections of future climate; however, their roles are so complex that uncertainty in radiative forcing of climate change is mainly dominated by the uncertainty associated with aerosol forcing (Forster et al., 2007). This complexity is due to aerosols' role in altering the planetary energy balance through a number of mechanisms: direct effects (Haywood and Boucher, 2000), semi-direct effects (Hansen et al., 1997) and indirect effects (Lohmann and Feichter, 2005).





The lack of detailed knowledge of the emissions, optical and chemical properties of aerosols results in knowledge
gap that prevents a full understanding of aerosol impact on climate simulations (Ghan et al., 2012).

3       In contrast to climate models, global Numerical Weather Prediction (NWP) centers have used monthly
climatologies of aerosol distributions to account for aerosol effects in the past. This is largely due to the additional
complexity and computational resources required to include fully prognostic aerosol schemes in high-resolution
operational global forecasting systems, but is is also due to a limited understanding of aerosol feedbacks in short-
range (1-5 day) forecasts. However, the advancement in computing power, improved aerosol models, and enhanced
aerosol observations now allow a more systematic documentation of the impact of aerosols (and uncertainties
therein) on weather forecasts (Tanaka et al., 2003; Morcrette et al., 2009, Westphal et al., 2009). Some of the NWP
centers have embarked on aerosol data assimilation efforts using both passive and active sensors (Tanaka and Chiba,
2005; Zhang el al., 2008; Benedetti et al., 2009). Several studies have shown improvement in NWP forecasts by the
inclusion of aerosols (Haywood et al., 2005; Mulcahy et al., 2014). Short range forecasts of aerosols by NWP
centers are particularly beneficial for air quality forecasts and other societal needs in the event of large dust events
(like trans-Atlantic dust plumes from Sahara) or biomass burning episodes (e.g., Southern Africa, North and South
America and South-East Asia).
Verification of aerosol forecasts against available observations is important to correct systematic model
biases and to understand the model's variability characteristics. Previous studies have been done evaluating the
performance of the European Centre for Medium Range Forecast (ECMWF) aerosol model by comparing model
data against satellite and ground observations (Morcrette et al., 2009; Mangold et al., 2011; Cesnulyte et al., 2015).
These studies focused on the comparison of monthly mean and daily aerosol quantities in both visible and UV
wavelengths as well as looking into different case studies (e.g., Saharan dust event, high sea-salt aerosol load, etc.).
Eskes et al., 2015 provided a general overview of the validation approach for the European MACC (Monitoring
Atmospheric Composition and Climate) global forecast system which uses data assimilation to combine in-situ and
remote sensing observations for atmospheric aerosols. Campbell et al. (2012) evaluated NASA Cloud Aerosol Lidar
with Orthogonal Polarization (CALIOP) aerosol optical thickness (AOT) against the Navy Aerosol Analysis and
Prediction System (NAAPS) to qualitatively assess day/night retrieval skill of the satellite and its accuracy. NAAPS
also developed an AOT reanalysis product using the assimilation of quality controlled retrievals from the satellite
and found the reanalysis follows the seasonal and interannual variability for the total AOT quite well (Lynch et al.,
29  2016).

30       At NOAA, a prognostic aerosol capability was developed at the Environmental Modeling Centre (EMC) of
the National Centers of Environmental Prediction (NCEP) in 2012. NASA's bulk aerosol scheme (an in-line version
of the Goddard Chemistry, Aerosol, Radiation and Transport model [GOCART], Chin et al., 2002, Colarco et al.,
2010) was incorporated into the NOAA Environmental Modeling System (NEMS) to establish an interactive global
aerosol forecasting system, NEMS GFS Aerosol Component version 1.0 (hereafter NGACv1) (Lu et al., 2016). The
model became operational in 2012, providing 120-hour global dust forecasts, once per day. It was incorporated as
one of the seven global models in the world's first global multi-model aerosol ensemble product, the International





Cooperative for Aerosol Prediction Multi-Model ensemble: ICAP-MME (Sessions et al., 2015) to forecast dust in real-time basis. NGACv1 was also incorporated into the World Meteorological Organization (WMO) Sand and Dust Storm Warning Advisory and Assessment System (SDS-WAS) Northern Africa-Middle East-Europe (NA-ME-E) node to provide timely and quality sand and dust storm forecasts.

NGACv1 was recently upgraded to include four more aerosol species (sea-salt, sulfate, black carbon, and organic carbon) from its previous version of dust-only forecasts. This upgrade of the model (hereafter NGACv2) also uses near-real time satellite based smoke emissions and was declared operational in March 2017. The focus of this paper is the evaluation of the NGACv2 AOT product at 550nm. The paper is organized as follows: Section 2 presents general information about the NGAC model and a summary of the products. Satellite and ground data sets used in this evaluation are described in Section 3. Section 4 shows comparisons of NGACv2 with ICAP-MME and satellite retrievals. The evaluation of NGACv2 aerosol products with in-situ measurements is presented in Section 5. Section 6 describes two events (one is Central African smoke and the other is Trans-Atlantic dust) where NGAC forecasts are compared against observations. Section 7 finishes with a discussion and concluding remarks. Detailed descriptions about NGACv2 and its outputs, and its operational implementation are described in Part 1 of this paper (Wang et al., 2017).

## 2. Model Description

NGACv2 is a global in-line aerosol forecast system. The forecast model component of NGAC is NOAA's operational Global Forecast System (GFS) based on NEMS, which, in turn, is based on the common modeling framework using the Earth System Modeling Framework (ESMF). GFS is a spectral model, comprised of model dynamics and physics in a hydrostatic system with a reduced Gaussian grid and hybrid (sigma and pressure) vertical levels. The aerosol component of NGACv2 is GOCART, which was developed at NASA Earth Science Programs to simulate atmospheric aerosols (including sulfate, black carbon (BC), organic carbon (OC), dust and sea-salt) and sulfur gases ($SO_2$) (Chin et al., 2002, 2007; Ginoux et al., 2001; Colarco et al., 2010). Dust and sea-salt emissions are dependent on wind speed, whereas BC and OC are produced from biomass burning and biofuel consumption. Sulphate is produced from the oxidation of $SO_2$ and dimethylsulphide (DMS). Daily biomass burning emissions are provided by the Global Biomass Burning Emission Product extended (GBBEPx) which was developed at NOAA's National Environmental Satellite, Data and Information Services (NESDIS) Center for Satellite Application and Research (STAR). GBBEPx contains daily global biomass burning emissions (BC, OC, $SO_2$ etc.), blended fire observations from NESDIS/STAR's Global Burning Emission Product from a constellation of Geostationary satellites (GBBEP, Zhang et al., 2012) and NGAC/GMAO's Quick Fire Emissions Data version 2 from a polar orbiting sensor (QFED2, Darmenov and Dal Silva, 2015). NGACv2 is a joint collaboration between NOAA & NASA and represents an efficient way of transitioning research into NCEP operations. More details about model configuration, emission data sets, budget, post-processing and NEMS GFS coupling with GOCART are discussed in Wang et al., 2017.



NGACv2 currently runs at T126L64 (~110km) which is a lower horizontal resolution than the current
operational GFS (T1534L64, ~13km as of March, 2017). Aerosol initial conditions are taken from the 24–hour
NGAC forecasts from the previous day while meteorological initial conditions are down-scaled from the high-
resolution Global Data Assimilation System (GDAS) analysis. NGACv2 runs twice a day at 00z and 12z  and
produces output on 1°x1° degree longitude/latitude grid at 3-hourly forecast intervals from 00 to 120 hours. Output
files contain both 2-dimensional and 3-dimensional fields of various aerosol and meteorological variables. Total
AOT is calculated based on all 5 species of aerosol at 340, 440, 550, 660, 860, 1110 and 1630nm wavelengths. AOT
from each species at 550nm is also available, apart from mixing ratios (in 3-dimensions), sedimentation flux, dry
and wet deposition flux and scavenging flux. A full list of NGACv2 output is available at Wang et al., 2017.
**3. Data**
Here we describe both NGACv2 and other observational AOT datasets used in this study. As AOT (column
integrated extinction coefficient) at 550nm is a common reference for much of the previous work that involves
satellite aerosol retrievals, we have considered this one quantity for all the evaluations. Daily NGACv2 forecast data
from June 2015 to October 2016 (17 months total) is used to evaluate spatial and temporal variation for global and
regional scales. NGACv2 550nm AOT (total and individual species) data is two dimensional (1°x1° degree grid) and
in GRIB2 format.
MODIS provides near-global coverage of aerosol measurements in space and time. We used a MODIS
Level-3 (daily and monthly at 1°x1° degree) AOT dataset in this study (https://ladsweb.nascom.nasa.gov/). The
dataset belongs to the Collection 6 combined land and ocean from the Aqua satellite (Levy et al., 2013). This latest
collection of MODIS data includes AOT data based on refined retrieval algorithms, in particular the expanded Deep
Blue algorithm (Hsu et al., 2013; Sayer et al., 2013). It introduces a merged AOD product, combining retrievals
from the Dark Target (DT) and Deep Blue (DB) algorithms to produce a consistent data set covering a multitude of
surface types ranging from oceans to bright deserts (Sayer et al., 2014). We have used 550nm MODIS AOT
variables "dark target" and "deep blue" (for brighter surfaces) for all the statistical comparisons in this paper. We
also used the new aerosol product "Dark_Target_Deep_Blue_Combined_Mean" to qualitatively compare model
results.
The Visible Infrared Imaging Radiometer Suite (VIIRS) sensor onboard the Suomi National Polar Orbiting
(SNPP) satellite provides sets of aerosol Environmental Data Records (EDRs) based on daily global observations
from space (Jackson et al., 2013, Liu et al., 2013). Beginning in 2012, VIIRS provides AOT at 550nm at a global
0.25°x0.25° horizontal resolution. Daily gridded VIIRS data used in this paper are from the NOAA STAR ftp site at
ftp://ftp.star.nesdis.noaa.gov/pub/smcd/jhuang/npp.viirs.aerosol.data/edraot550. We also have used Enterprise
Processing System (EPS) VIIRS data (1°x1° resolution), which uses a newer aerosol algorithm to retrieve AOT for a
dust event in Africa (Ciren et al., 2012; Laszlo and Liu, 2016) and became operational in July 2017.
ICAP-MME provides 6 hourly forecasts of total and dust AOD globally out to 120 hours at 1°x1° degree
resolution (Reid et al., 2011, Sessions et al., 2015). Total AOD in ICAP-MME is provided by the four core multi-



species models: the European Centre Medium Range Weather Forecasts Monitoring Atmospheric Composition and Climate Model (ECMWF-MACC), Japan Meteorological Agency Model of Aerosol species in the Global Atmosphere (JMA-MASSINGAR), NASA Goddard Earth Observing System Version5 (NASA-GEOS5) and Naval Research Lab Navy Aerosol Analysis and Prediction System (NRL-NAAPS) modeling systems. Dust-only AOD are provided by the aforementioned four models, plus the Barcelona Supercomputer Center Chemical Transport Model (NMMB/BSC-CTM), United Kingdom Met Office Unified Model (UKMO) and NGACv1. All four of the multi-species models invoke aerosol data assimilation (DA) and satellite-based smoke emissions. In this study, we have used coincident 6-hourly ICAP-MME forecasts of each day to compare against NGACv2 results. Multi-model ensembles, which use independent and skilled forecasts, are an ever increasing tool for forecasters as they are more accurate than the individual member deterministic models (Meehl et al., 2007; Fordham et al., 2012). As NGACv2, ICAP-MME and MODIS products all have 1° horizontal resolution; no horizontal interpolation was needed to put the different data sources onto a single grid.

The Aerosol Robotic Network (AERONET) is a global ground-based network of automated sun-photometer measurements that provide AOT, surface solar flux and other radiometric products (Holben et al., 1998). It is a well-established network of over 700 global stations and its data are widely used for aerosol related studies (Zhao et al., 2002). AERONET employs the CIMEL sun-sky spectral radiometer which measures direct sun radiances at eight spectral channels centered at 340, 380, 440, 500, 675, 870, 940 and 1020 nm. AOT uncertainties in the direct sun measurements are within ±0.01 for longer wavelengths (longer than 440 nm) and ±0.02 for shorter wavelengths (Eck et al., 1999). To compare with NGACv2 550nm AOT data, AERONET AOT at 440 nm and 675 nm were linearly interpolated on a log-log scale to provide 550nm AOT. All AERONET data are sampled temporally at ±1 hour of daily 3-hourly NGACv2 forecasts (for example, at any particular location AERONET measurements between 11z and 13z are averaged to compare against the 12z model forecast). A 2-hour time window is created to allow for more sampling of AERONET measurements over any location. Also, we discarded very high AERONET AOT values (over 2.5) from all stations when statistical analysis was performed. Model AOT at a site was extracted and compared only when AERONET had measurements in that time window. In this study we have used all available level 1.5 (cloud screened) daily AOT data sets for the same time period (Smirnov et al., 2000).

Quantitative analysis in this study is performed by calculating the following parameters: the average, standard deviation, correlation coefficient (R) and root mean square error (RMSE) of unitless 550nm AOT.

**4. Comparison with Satellite Observations and ICAP-MME**

We compared seasonal variations (all four seasons: JJA, SON, DJF and MAM) of model forecast AOT with MODIS data for 2015-16. Figure 1 shows global maps of AOT (total and dust from NGACv2) against ICAP-MME and MODIS (total AOT) for 2005 JJA (average of June-July-August). Higher burdens of AOT are found during the Northern Hemisphere summer, as wind-blown dust over northern Africa and the Persian Gulf and smoke over southern Africa and Northern America contributes the majority of high AOT shown in Figure 1. NGACv2 seasonal



variation is in qualitative agreement with both MODIS and ICAP-MME for many of the locations that represent
major aerosol regimes, although there are a few noticeable differences. Major dust events over Africa, the Middle
East and north-western China are very similar in dust-only AODs between NGACv2 and ICAP-MME (Figure 1b,
d). Dust transported plumes from northern Africa to the Atlantic Ocean are the most visible feature for both the
models and satellite products (Figures 1a, c, and e). Smoke events located on the western coast of South Africa and
Canada are from NGACv2 OC AOT (not shown). Persistent sea-salt aerosol bands at 60ºS are evident from model
total AOT (Figures 1a, c). Some of the differences in total AOT (for example, lower AOT over India and China) are
the results of known issues associated with NGACv2 which will be discussed later on. Since daily gridded MODIS
data are used, which are not sampled at model forecast hours, some of the differences between NGACv2 and
MODIS can be attributed to data sampling.
Figure 1 showed that Saharan dust dominates most of the observed high AOT in the atmosphere over the
Atlantic Ocean in the summer months. We also analyzed monthly variations of meridional distributions of AOT
over the Atlantic Ocean. Figure 2 shows NGACv2 total, dust and OC AOT between 40ºS to 60ºN in three different
months, December 2015, and April and July 2016. In the Hovmöller diagrams (Figure 2), 6-hourly model forecasts
are averaged between 60ºW and 30ºE (including land regions over Africa and Europe) to get daily AOT values from
the model for each month. We also plotted latitudinal variation of AOT from our model at a 23ºW longitude transect
(located over the Atlantic Ocean where the majority of the aerosol plumes pass) for the same months and aerosol
species (line plots in Figure 2). We added MODIS total AOT at 23ºW to validate our model results. Latitudinal
changes in the aerosol plume off the coast of western African coast are shown by NGACv2 for the selected months.
In the winter (Figures 2 a-c) maximum values of AOT are located around 10ºN, but in July the max moves further
north to around 18-20ºN (Figures 2 g-i). Biomass burning in northern Africa is most active in the winter season, as
OC AOT shows high values between 0-5ºN (Figure 2c). So, high values of total AOT in Figure 2a are contributed to
by dust and OC aerosols and also by sea-salt aerosols in higher latitudes between 50º-60ºN (not shown). In contrast,
in July the total AOT peak shifts to 20ºN (Figure 2g) and dust is the dominant aerosol contributing to total AOT
(Figure 2h). In the summer season dust originates from the western Sahara, under the conditions of a thermal low
that prevails over that region (due to intense solar heating). In July 2016 biomass burning contributed much of OC
aerosols across the Atlantic south of the equator (Figure 2i). Also, OC and sulfate (not shown) from Europe
contributes to total AOT in July (Figure 1g). Compared to strong latitudinal variations in December and July, all the
AOT peaks are less intense in April (Figures 2d-f), with the majority contribution from dust aerosols. Model results
agree with latitudinal variation at the 23ºW location, where total AOT peaks match between NGACv2 and MODIS
across all three months (Figures 2a, d and g). Seasonal shifts of trans-Atlantic aerosol plumes are of the s kind that
have been observed through satellites and reported in numerous studies (Takemura et al. 2000; Kaufman et al. 2005;
Ben-Ami et al., 2009).
For quantitative comparisons we selected key aerosol regions over the land and ocean, and extracted the
model results and satellite data over those regions (Figure 3). We have used 6-hourly model forecasts and averaged
them to calculate the daily mean AOT values over these regions. The three ocean regions include the North and





South Atlantic Oceans and North Indian Ocean, which are major long-range aerosol transport pathways for dust,
smoke and sulfate. Figure 3 shows nine land regions including two dust source regions (North Africa and the Middle
East), two biomass burning regions (South America and South Africa), three regions over North America (eastern
and western US and Canada) and two major pollution source regions (India and East Asia). Previous studies have
shown that aerosols over India and East Asia are composed of different aerosol types and the relative contribution of
individual species varies with season (Kedia et al., 2014; Bhawar et al., 2016). Table 1 summarizes the latitude-
longitude bounds of all the twelve regions, along with correlation coefficients and RMSEs for NGACv2 and MODIS
for different seasons between 2015 and 2016. Figure 4 shows one such daily time-series for 2015-JJA in the selected
six regions where we have included ICAP-MME results as well. The time series of individual regions provides a
general characterization of the overall difference between model and satellite products. Figures 4a and c show
NGACv2 agrees very well with both ICAP-MME and MODIS over one strong biomass burning event in Northern
America during late June-early July of 2015. For African dust, NGACv2 correlates well when the dust plume is
present over land (Figure 4e), but underestimates it over the ocean (Figure 4d).
Over the oceans, the model shows consistently high correlations with MODIS across different seasons
(Table 1). Both the North and South Atlantic Oceans are dominated by trans-Atlantic passages of dust, smoke (both
BC and OC) and aerosol plumes as well as the presence of sea-salts. On the other hand, dust from Arabian Peninsula
travels across the northern Indian Ocean between May to August to reach the Indian subcontinent (Shalaby et al.,
2015). In the winter, pollution outflow from the Indian subcontinent creates a haze plume over the ocean
(Ramanathan et al., 2001). NGACv2 shows low RMSE error (and high correlations) in both the North and South
Atlantic Ocean. However, higher RMSE is associated over the Indian Ocean during both summer seasons and that is
related to an underestimation of dust transport from Middle East.
Over land, the performance of NGACv2 is mixed across different regions, as shown in Table 1. Over the
continental US (both eastern and western US), the model shows both high correlations (more than 0.5 in all seasons,
except the summer of 2016) and low RMSE (less than 0.12) compared to satellite products in all six seasons of the
current analysis. We noticed a drop in the correlation coefficient in summer 2016 (0.41) from the previous summer
(0.66) in the eastern US (RMSE remains low in both summers) and that can be partly due to the absence of a very
high aerosol event (Canadian smoke event) like the one that occurred in 2015 (Figures 4a,b). In summer 2016, the
highest total AOT averaged over the eastern US from MODIS is 0.35, compared to 0.78 in 2015. The modeled and
MODIS AOTs in the Saharan dust source region (N. Africa) show a correlation over 0.6 (with low RMSE) during
the major dust outbreak seasons in summer. Over the biomass burning regions (in South America), the model shows
low correlation (and high RMSE) during September-November, when most of the Amazon forest fires take place.
But in the non-burning season both the correlations and RMSE improve. The magnitude of the maximum AOT over
South America is largely underestimated by the model by a factor of 3, indicating that the biomass burning emission
in the model is probably too low during the burning season.
One major difference between the model and the satellite data is over India, where the model has a much
lower AOT in all seasons (low R and high RMSEs). The largest contribution from aerosol loading over India comes



from the anthropogenic component (with the majority as sulfate, followed by OC and BC) and by dust blown from the Middle East and western India during May-July. This bias in AOT by NGACv2 may be due to high aerosol scavenging by clouds and precipitation and their subsequent removal of them from the atmosphere. Also, dust blown from Middle-East is underestimated by NGACv2 (Figure 4f) contributing to lower AOT in the pre-monsoon season over India. Yoo et al. (2013) evaluated GFS forecasts against satellite observations and identified large discrepancies in low cloud fractions over land and oceans. There could be several factors responsible for such discrepancies, such as a) removal of cloud condensate water by strong vertical diffusion in the shallow convective scheme, b) microphysical processes interacting with strato-cumulus clouds can remove cloud condensate water c) the precipitation scheme used in the model leads to large aerosol removal through wet deposition. The GFS also tends to overestimate cloud layer thickness, particularly for deep convective clouds in the tropical regions. All this could cause the low bias in AOT over India (and East Asia) as sulfate aerosols (and also 20% BC and 50% OC in GOCART are hydrophilic) are formed in the clouds and hygroscopic growth is most effective in high humidity regions near clouds.

## 5. Comparison with AERONET

Figure 5 shows correlation coefficients (R) of the NGACv2 AOT compared to AERONET derived AOT during the entire 17 months of the study period. Table 2 summarizes the latitude, longitude of the AERONET sites along with R, RMSE, and number of paired observation points of the 57 stations used in this study. Figure 6 shows a scatter plot of 550nm total AOT between NGACv2 and AERONET at twelve stations. The first seven sites in Figure 6 are located on the west coast of northern Africa and are dominated by dust aerosols. The model closely reproduces the observed variation (with R between 0.5-0.6 and low RMSE). Site 8 (Tamanrasset), located at the center of the Sahara Desert, shows very high R (0.74) because of its location in the active dust source area (maxima of the dust source function in the model) (Figure 6a). However, the model overestimates AOT during the low dust AOT period (November to March) over this site which leads to higher RMSE.

Sites 9-12 are located at the northern boundary of Africa, and are influenced by dust from the Sahara: Oujda in Morocco, Graciosa island in the Azores (in the Atlantic Ocean), Tizi Ouzou in Algeria and Ben Salem in Tunisia (Figure 6e). These sites are located further from the Sahara (compared to the first six sites), but the transport of dust simulated by the model matches closely with observations (with R ~0.5). Aerosols at sites 13-17 in Figure 5 contain dust aerosols from Africa and other aerosol types from the European landmass. All these sites are located in southern Europe (near the western part of Mediterranean Sea) and are influenced by desert dust transported from arid areas in North Africa and advection of anthropogenic particles from the central European industrial area (Mallet et al., 2013). Table 2 and Figure 6f suggest model AOT correlations vary between 0.32-0.62 at these sites, with associated low RMSE.

Sites 18 and 19 are located in the Middle East and consist mainly of mineral dust. NGACv2 correlates better with the King Abdullah University of Science and Technology (KAUST) campus site (located in Saudi Arabia) with a correlation above 0.6, but the correlation decreases to 0.52 at Sede Boker, which is located further





north on the Arabian Peninsula. Despite a high correlation at the KAUST campus site, the model often underestimates some of the higher AOT events at this location, which gave rise to a higher RMSE (~0.32). Sites 21-23, located in equatorial and southern Africa, are influenced mainly by biomass burning. Biomass burning activity peaks during August-September at these sites and the magnitude of the maximum AOT at the three South African sites is underestimated by the model by a factor of almost 2 to 3 times (high RMSE in Table 2), suggesting that the biomass burning emission in the model is probably low during the burning season. Similar underestimation of AOT is also observed over two of the South American sites (24 and 25 in Figure 5). Model simulated AOT correlates well (0.58) at site 24, which is largely due to the model estimating low AOT at these sites during the non-biomass burning seasons (Figure 6g). But the model underestimates AOT (~3 times) between September-November when the biomass burning season prevails in Brazil.

Site 20 and sites 26 to 44 in Figure 5 are located in and around North America (US and Canada) and are generally considered to be dominated by pollution aerosols: smoke, sulfate and dust in the south eastern and southwestern US. Three sites (26, 38 and 40 in Table 2), which are in Canada and located above 55ºN, are influenced by biomass burning aerosols and trans-Pacific transport of pollutants (mainly dust). All three sites show higher correlation with the model (R above 0.4) and the model closely reproduces the higher AOT over Fort McMurray and Yellowknife during major fire events that included May 2016 around Fort McMurray. The rest of the locations over the continental US (hereafter CONUS) show mixed results in terms of R and RMSE (Table 2). South western sites (sites 36, 43 and 44) influenced by dust in the spring and sulfate in summer show R of around 0.4. Sites in the northern and north eastern parts of CONUS are dominated by anthropogenic pollution (sulfate) and occasional smoke from Canada in winter and spring. NGACv2 correlates reasonably well with (R ~0.35) model underestimation of sulfate aerosols in summer. Also, the model does not have nitrate aerosols from anthropogenic sources, which leads to underestimation of AOT. Kroll and Seinfeld (2008) have shown that anthropogenically emitted nitrogen oxides ($NO_x$) can directly affect the formation of secondary organic aerosols (SOA).

The rest of the thirteen sites (Sites 45-57 in Figure 5 and Table 2) are located all over the globe reflecting a variety of aerosol regimes. For example, at the oceanic site in Hawaii (Site 46 in Figure 5), modeled AOT values are higher than AERONET between May to October. This bias could be due to overestimation of trans-Pacific dust transport from Asia and sea-salt aerosols. A similar overestimation of Asian dust is also observed at Dalanzadgad (site 51) which is located in the arid Gobi desert region in Mongolia. Over urban areas (sites 45, 47, 48) model correlations with AERONET are moderate (R ~0.3) with an underestimation of AOT in summer over Mexico City (site 45) and Kyiv (site 47). Ascension Island (site 49) is located in the remote southern Atlantic Ocean and is affected by biomass burning outflow from southern Africa (Figure 6k). The model is able to reproduce high biomass burning events over this location as shown by a high correlation (R=0.55) and low RMSE (Table 2). Sea-salt aerosol is dominant over remote Amsterdam Island in the southern Indian Ocean and model correlation is moderate (R=0.28) but with low RMSE. NGACv2 shows R ~0.32 with AERONET measurements at three larger metropolitan cities (sites 52, 54 and 55 in Table 2), with an underestimation of sulfate and anthropogenic aerosols during the summer months at all three Asian locations (Figure 6i).



## 6. Case Studies

### 6.1 July 2016 Smoke event

Forest fires are a significant source of carbonaceous aerosols at northern latitudes in spring and summer (Generoso et al., 2003) and are associated with increased mortality and morbidity (Rappold et al., 2011). A major fire breakout was reported in central Africa during July and August 2016. The majority of the fires burned cropland or grass, which is a common agricultural practice in this region. We compared model forecasts and observations on selected days in July over this region to assess model performance during this event. Figure 7 shows a comparison of the total AOT between NGACv2, ICAP-MME, VIIRS and MODIS for days when smoke emission is prominent. We have averaged coincident 6-hourly model forecasts (for both NGACv2 and ICAP-MME) to compute daily averages to compare against daily satellite observations. 10m zonal wind from NGACv2 (not shown) indicates an easterly wind gradually pushed smoke from Central Africa towards the west and north west in the month of July. Figure 7 shows NGACv2 captured this smoke event quite well and qualitatively matches (in terms of location and advection) with both ICAP-MME and satellite observations. The magnitude of AOT however is underestimated by the model compared to ground station and satellite observations (Figure 7 and 8). Smoke AOT has been added as a new capability in NGACv2 and uses different emissions than the models that are under the ICAP assembly, which independently verifies model performance.

We also looked into model AOT against one AERONET station in Central Africa (Figure 8) during this fire event. The location of that AERONET station (station "SEGC_Lope_Gabon" in AEORNET database) is marked in Figure 7. We compared total, OC and BC (only) AOT from NGACv2 against observed AOT at that station. Both model and station observations show an increase in AOT after July 10[th], which continues to grow higher after July 15[th] until the end of the month. The majority of total AOT in Figure 8 is contributed by biomass burning generated OC, with some increase in BC also observed. Figure 8 shows the model AOT pattern for the month matches closely with surface observations. In terms of intensity, some of the reported AOT from AEORNET are higher than the model forecast, which is also due to a difference in spatial resolution between the model and surface observations.

### 6.2 June 2015 Dust event

During boreal summer, dust from the deserts of the Sahara, the largest sources of dust in the world, is transported across the Atlantic Ocean by prevailing tropical easterly winds (Karyampudi et al., 1999). According to recent satellite estimation, each year 182 million tons of dust on average gets past the western edge of Sahara and out of that 27.7 million tons fall on the surface of the Amazon basin (Yu et al., 2015). Huge plumes of Saharan dust swept off the coast of Western Sahara in the middle of June 2015 and traveled across the Atlantic Ocean to reach the southeast corner of the US (UMBC smog blog reported days of dust in the Caribbean and Gulf of Mexico at http://alg.umbc.edu/usaq/archives/2015_06.html). The actual dust storm began on June 13[th] when a storm system off of the west coast of Africa kicked up a heavy stream of dust from Senegal, Western Sahara and Mauritania. On June 22[nd], the Saharan dust had traveled more than 5,000 miles to reach southern Texas, where it contributed to moderately poor air quality. Figure 9 shows NGACv2 total AOT forecasts for the selected days of June 13[th], 17[th]





and 21st. These days show the progression of dust westward from the African coast with high AOT above 1 over
land which gradually decreases as the dust storm crosses over the ocean. ICAP, MODIS and EPS-VIIRS (all in 1° x
1° horizontal resolution) are compared against NGACv2 in Figure 9.
Four AERONET stations (marked in the first figure in the Figure 9 panel) were used in this case to further
look into the westward dust progression. One of these four stations, Tamanrasset (22ºN, 5ºE) in southern Algeria, is
located near the source of dust storm, while other three stations, Capo Verde, Cape San Juan and Guadeloupe, are
located on the downwind side. Total AOT from AERONET are compared against total, dust and OC AOT from
NGACv2 in Figure 10 for each of these four stations. It is evident that dust AOT is the main contributor to total
NGACv2 AOT at all the stations during this event. Between June 8th and 21st, the AERONET location in
Tamanrasset observed ground AOT above ~0.7 on some days with highs reaching nearly 1.5 (Figure 10a). Apart
from on June 8th, NGACv2 dust AOT intensity (reaching ~0.6) was underestimated compared to ground
observations at this location. At Capo Verde, which is located just off the coast of Africa, NGACv2 correlates well
with AERONET observations, but overestimates the intensity (nearly 2 times) during the event (Figure 10b). San
Juan and Guadeloupe stations, located in Puerto Rico and the Caribbean respectively, show a gradual increase in
AOT from June 13th onward as Saharan dust began to reach those locations (Figures 10c & d). NGACv2 dust AOT
peaks coincide with high AERONET values at these locations.
**7. Summary and Conclusions**
This paper presents an evaluation of NOAA's new updated aerosol forecast model NGACv2 which became
operational in March 2017. The model couples NEMS GFS with NASA's GOCART aerosol and is an in-line global
aerosol forecast system. The model forecasts five species of aerosol (dust, sea-salt, BC, OC and sulfate) every 3-
hours, twice per day (00z and 12z) and out to 5 days on a global 1°x1° horizontal grid. We extensively evaluated 17
months of model simulated total AOT both temporally and spatially against satellites (MODIS, VIIRS) and multi-
model ensemble (ICAP-MME) data. Satellite AOT retrievals inherently have greater uncertainty which is further
exacerbated by using measurements from multiple satellites. The long-term MODIS AOT on the other hand
provides a consistent measurement platform and hence it is used for the validation of model results in this study. We
also compared model results with more than 50 AERONET station observations, which are spread globally and
represent different aerosol regimes.
The model reproduces the prominent temporal and geographical features of AOTs as observed by MODIS
and ICAP-MME, like dust plumes over northern Africa and the Arabian Peninsula, biomass burning plumes in
southern Africa, northern Canada and high altitude sea-salt bands. The AOT in North Africa is among the highest in
the world throughout the year, a combined effect of dust outbreaks from the Sahara Desert and biomass burning near
the equator. NGACv2 captures the seasonal shift of the aerosol plume off the west coast of Africa and agrees well
with MODIS observation. The model also correlates highly with MODIS observations over both the eastern and
western US regions during the study period. We found an underestimation of model AOT over Asia, and during the
South American biomass season and Middle East dust season. We regularly monitor dust and smoke events around



the globe and use them to evaluate our model performance. In this paper, we showed two such cases, where the
NGACv2 forecast fared reasonably well against other models and observations with some biases in terms of
intensity.
The comparisons of model forecasts with surface point locations show results similar to our comparisons
against MODIS in larger gridded domains. The model reproduces the seasonal variations at most of the sites,
especially those sites on the African continent where dust and biomass plumes dominate. The model also captures
dust and smoke outflow from Africa at AERONET locations that are present in the Atlantic Ocean (Capo Verde,
Ascension Island) even though the magnitudes do not match with these point observations. Model AOT captures
two other dust regions: the Arabian Peninsula and Asian dust near the source region, but underestimates them
quantitatively as these dust plumes undergo long-range transport over Asia. The model forecasts large biomass
burnings over Canada in both 2015 and 2016 and it agrees well with AERONET station data. However, like other
aerosol forecast models, NGACv2 also produces weaker AOT signals for some aerosol events and regimes. The
model underestimates AOT over the Amazon region in both years and also for the Indonesian fire event in 2015. It
also underestimates sulfate AOT over Asia which results in at large underestimation of total AOT compared to
AERONET over these locations. At present, model comparisons with satellite results can be meaningfully
interpreted in regions where AOT is very high and dominated by a single aerosol (dust or smoke). In mixed aerosol
regimes, particularly over land (where pollution, long range transport of biomass burning or dust all contribute), the
model seems to simulate AOT lower than the observations by a factor of 2-3 times. We discussed some of these
problems associated with the model that include quick removal of aerosols (scavenging), the type of microphysics
scheme (creation of too few or excessive boundary layer clouds that reduce sulfate AOT generation) and lower
emission factors (over South America). Our next steps will be addressing these issues with the model and to further
improve overall model forecasts, with a particular focus on Asia. Ongoing DA work with NGAC shows some
improvements in terms of total AOT over Asia through DA (Lu et al., 2017).
Expanding the aerosol species from dust only in NGACv1 to multi-species in NGACv2 provides a more
complete global aerosol forecast using near-real-time global biomass burning emission data GBBEPx. It also
provides direct guidance on long-range aerosol transport and the impact on particulate matter over CONUS, and will
be used as the dynamical boundary conditions to a regional air quality model like the Community Model for Air
Quality (CMAQ) which runs as part of NOAA's National Air Quality Forecast Capability (Lee et al., 2017). This
work provides general validation results to characterize the present NGACv2 performance and identify deficiencies
for    future    improvements.    Daily    NGACv2    web    graphics    can    be    viewed    at
http://www.emc.ncep.noaa.gov/gmb/NGAC/html/realtime.ngac.html  and  near-real-time  comparisons  with  other
models, satellites and AERONET stations are posted at http://www.emc.ncep.noaa.gov/gmb/NGAC/NGACv2.
**Data and code availability**
NCEP operational products are accessible to general users, free of charge in real-time at NOAA Operational Model
Archive and Distribution System (NOMADS). NCEP Central Operations (NCO) ftp site provide the source code,





relevant run scripts and fixed fields files at http://www.nco.ncep.noaa.gov/pmb/codes/nwprod/ngac.v2.3.0. The NGACv2 output is in GRIdded Binary Version 2 (GRIB2) format on $1^0 \times 1^0$ degree grid, with 3-hourly output up to 120 hours. NGACv2 products from NOMADS are available at http://nomads.ncep.noaa.gov/pub/data/nccf/com/ngac/prod. NCAR Command Language (NCL) program is used to generate all the figures in this paper.

**Acknowledgments**

Partial funding of this research provided by the NOAA National Weather Service Program Office (NWSPO) Next Generation Global Prediction System (NGGPS) award number NA15NWS4680008. We thank Atmosphere Archive & Distribution System (LAADS) Distributed Active Archive Center (DAAC), located in the Goddard Space Flight Center in Greenbelt, Maryland for providing MODIS Level-3 data. We acknowledge S-NPP/VIIRS science team for the high quality products. We thank Dr. Jeff McQueen who is lead in air quality forecast at NCEP/EMC for his support. We are grateful to Dr. Shobha Kondragunta who is co-lead to VIIRS aerosol algorithm team for providing EPS-VIIRS data. We wish to thank ICAP team for providing their MME AOT data. We acknowledge AERONET team for the production of the data used in this work.

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



5   **Table 1. Correlation coefficients and RMSE (in italics) of total AOT at 550nm in different seasons between NGACv2 and**

6   **MODIS over selected regions of the globe. Daily AOT from model and satellite data are considered for these calculations.**

| Regions (Latitude, Longitude) | JJA-2015 | SON-2015 | DJF-2016 | MAM-2016 | JJA-2016 | SON-2016 |
|---|---|---|---|---|---|---|
| *Over Ocean* | | | | | | |
| N. Atlantic Ocean (0-35ºN; 10ºW-80ºW) | 0.733 *(0.07)* | 0.803 *(0.06)* | 0.852 *(0.11)* | 0.504 *(0.1)* | 0.622 *(0.06)* | 0.71 *(0.07)* |
| S. Atlantic Ocean (0-35ºS; 40ºW:20ºE) | 0.644 *(0.16)* | 0.894 *(0.13)* | 0.524 *(0.13)* | 0.439 *(0.09)* | 0.664 *(0.12)* | 0.896 *(0.13)* |
| N. Indian Ocean (0-24ºN; 40ºE-100ºE) | 0.779 *(0.23)* | 0.445 *(0.2)* | 0.724 *(0.19)* | 0.305 *(0.17)* | 0.698 *(0.26)* | 0.688 *(0.21)* |
| *Over Land* | | | | | | |
| N. Africa (0-30ºN;18ºW:30ºE) | 0.756 *(0.04)* | 0.438 *(0.03)* | 0.283 *(0.13)* | 0.389 *(0.06)* | 0.611 *(0.06)* | 0.265 *(0.03)* |
| S. Africa (0-30ºS; 8ºE-35ºE) | 0.203 *(0.15)* | 0.139 *(0.19)* | 0.227 *(0.12)* | 0.255 *(0.11)* | 0.257 *(0.12)* | 0.208 *(0.17)* |
| E. USA (25ºN-48ºN; 68ºW-95ºW) | 0.666 *(0.11)* | 0.744 *(0.06)* | 0.821 *(0.03)* | 0.863 *(0.04)* | 0.414 *(0.08)* | 0.84 *(0.05)* |
| W. USA (25ºN-48ºN; 95ºW-125ºW) | 0.79 *(0.08)* | 0.74 *(0.03)* | 0.712 *(0.02)* | 0.86 *(0.04)* | 0.81 *(0.03)* | 0.71 *(0.03)* |
| Canada (48ºN-70ºN; 60ºW-160ºW) | 0.703 *(0.11)* | 0.45 *(0.08)* | 0.232 *(0.07)* | 0.296 *(0.09)* | 0.484 *(0.07)* | 0.205 *(0.07)* |
| S. America (0-35ºS; 35ºW-80ºW) | 0.704 *(0.05)* | 0.246 *(0.16)* | 0.183 *(0.17)* | 0.482 *(0.09)* | 0.29 *(0.06)* | 0.103 *(0.13)* |
| Middle East (10ºN-32ºN; 30º-70ºE) | 0.67 *(0.08)* | 0.873 *(0.08)* | 0.687 *(0.07)* | 0.589 *(0.06)* | 0.2287 *(0.12)* | 0.855 *(0.08)* |
| E. Asia (20ºN-48ºN; 100ºE-140ºE) | 0.656 *(0.14)* | 0.498 *(0.1)* | 0.618 *(0.15)* | 0.502 *(0.2)* | 0.603 *(0.19)* | 0.467 *(0.14)* |
| India (8ºN-35ºN; 68ºE-95ºE) | 0.319 *(0.33)* | 0.587 *(0.28)* | 0.164 *(0.3)* | 0.605 *(0.24)* | 0.109 *(0.36)* | 0.354 *(0.31)* |



2    **Table 2. Locations of AERONET stations, correlations, RMSE and number of paired observations with NGACv2**

| Locations | Latitude, Longitude | Correlation Coefficients | RMSE | N (Sample No.) |
|---|---|---|---|---|
| 1. Dakar | 14ºN,16ºW | 0.554 | 0.356 | 1430 |
| 2. Ilorin | 8ºN,4ºE | 0.628 | 0.449 | 944 |
| 3. Banizoumbou | 13ºN,2ºE | 0.547 | 0.345 | 1516 |
| 4. La Laguna | 28ºN,16ºW | 0.686 | 0.204 | 901 |
| 5. Saada | 31ºN,8ºW | 0.633 | 0.157 | 1575 |
| 6. Capo Verde | 16ºN,22ºW | 0.611 | 0.213 | 1089 |
| 7. IER Cinzana | 13ºN,5ºW | 0.565 | 0.293 | 1070 |
| 8. Tamanrasset | 22ºN,5ºE | 0.744 | 0.245 | 1333 |
| 9. Oujda | 34ºN,1ºW | 0.397 | 0.179 | 535 |
| 10. ARM-Graciosa | 39ºN,28ºW | 0.544 | 0.064 | 750 |
| 11. Tizi Ouzou | 36ºN,4ºE | 0.614 | 0.117 | 942 |
| 12. Ben Salem | 35ºN,9ºE | 0.681 | 0.144 | 1131 |
| 13. Barcelona | 41ºN,2ºE | 0.497 | 0.144 | 1123 |
| 14. Granada | 37ºN,3ºW | 0.62 | 0.122 | 1602 |
| 15. Mallorca | 39ºN,2ºE | 0.588 | 0.113 | 1273 |
| 16. Toulon | 43ºN,6ºE | 0.401 | 0.172 | 1025 |
| 17. Cabo da Roca | 38ºN,9ºW | 0.326 | 0.157 | 833 |
| 18. Sede Boker | 30ºN,34ºE | 0.522 | 0.146 | 1778 |
| 19. Kaust Campus | 22ºN,39ºE | 0.606 | 0.324 | 1601 |
| 20. Cape San Juan | 18ºN,65ºW | 0.578 | 0.14 | 822 |
| 21. SEGC Gabon | 0ºS,11ºE | 0.699 | 0.575 | 504 |
| 22. Mongu Inn | 15ºS,23ºE | 0.603 | 0.394 | 984 |
| 23. ICIPE Mbita | 0ºN,34ºE | 0.395 | 0.502 | 752 |
| 24. Alta Floresta | 9ºS,56ºW | 0.582 | 0.37 | 926 |
| 25. Manaus | 2ºS,59ºW | 0.303 | 0.415 | 769 |
| 26. Ft. McMurray | 56ºN,111ºW | 0.459 | 0.2611 | 582 |
| 27. Saturn Island | 48ºN,123ºW | 0.194 | 0.2 | 660 |
| 28. Bozeman | 45ºN,111ºW | 0.256 | 0.187 | 799 |
| 29. Halifax | 44ºN,63ºW | 0.409 | 0.177 | 955 |
| 30. Toronto | 43ºN,79ºW | 0.228 | 0.221 | 1066 |
| 31. Bondville | 40ºN,88ºW | 0.364 | 0.185 | 786 |
| 32. GSFC | 38ºN,76ºW | 0.301 | 0.18 | 1062 |
| 33. Key Biscayne | 25ºN,80ºW | 0.365 | 0.196 | 645 |
| 34. ARM- Cart site | 36ºN,97ºW | 0.327 | 0.1588 | 899 |



| 35. Trinidad Head | 41ºN,124ºW | 0.266 | 0.176 | 612 |
|---|---|---|---|---|
| 36. Tucson | 32ºN,110ºW | 0.415 | 0.107 | 882 |
| 37. Chapais | 49ºN,74ºW | 0.229 | 0.234 | 642 |
| 38. Yellowknife | 62ºN,114ºW | 0.4 | 0.251 | 513 |
| 39. Sioux Falls | 43ºN,96ºW | 0.422 | 0.212 | 658 |
| 40. Bonanza Creek | 64ºN,148ºW | 0.593 | 0.276 | 382 |
| 41. Georgia Tech. | 33ºN,84ºW | 0.372 | 0.175 | 689 |
| 42. LISCO | 40ºN,73ºW | 0.315 | 0.144 | 698 |
| 43. Fresno_2 | 36ºN,119ºW | 0.481 | 0.186 | 680 |
| 44. Sevilleta | 34ºN,106ºW | 0.392 | 0.054 | 891 |
| 45. Mexico City | 19ºN,99ºW | 0.248 | 0.429 | 599 |
| 46. Mauna Loa | 19ºN,155ºW | 0.053 | 0.073 | 1202 |
| 47. Kyiv | 50ºN,30ºE | 0.284 | 0.216 | 1086 |
| 48. Leipzig | 51ºN,12ºE | 0.233 | 0.182 | 830 |
| 49. Ascension Island | 7ºS,14ºW | 0.556 | 0.169 | 1257 |
| 50. Issyk-Kul | 42ºN,78ºE | 0.415 | 0.14 | 882 |
| 51. Dalanzadgad | 43ºN,104ºE | 0.308 | 0.164 | 626 |
| 52. Beijing | 39ºN,116ºE | 0.416 | 0.614 | 636 |
| 53. Chiayi | 23ºN,120ºE | 0.165 | 0.699 | 526 |
| 54. Jaipur | 26ºN,75ºE | 0.293 | 0.432 | 917 |
| 55. Karachi | 24ºN,67ºE | 0.358 | 0.345 | 816 |
| 56. Hanimaadhoo | 6ºN,73ºE | 0.184 | 0.409 | 995 |
| 57. Amsterdam Isld. | 37ºS,77ºE | 0.284 | 0.111 | 523 |





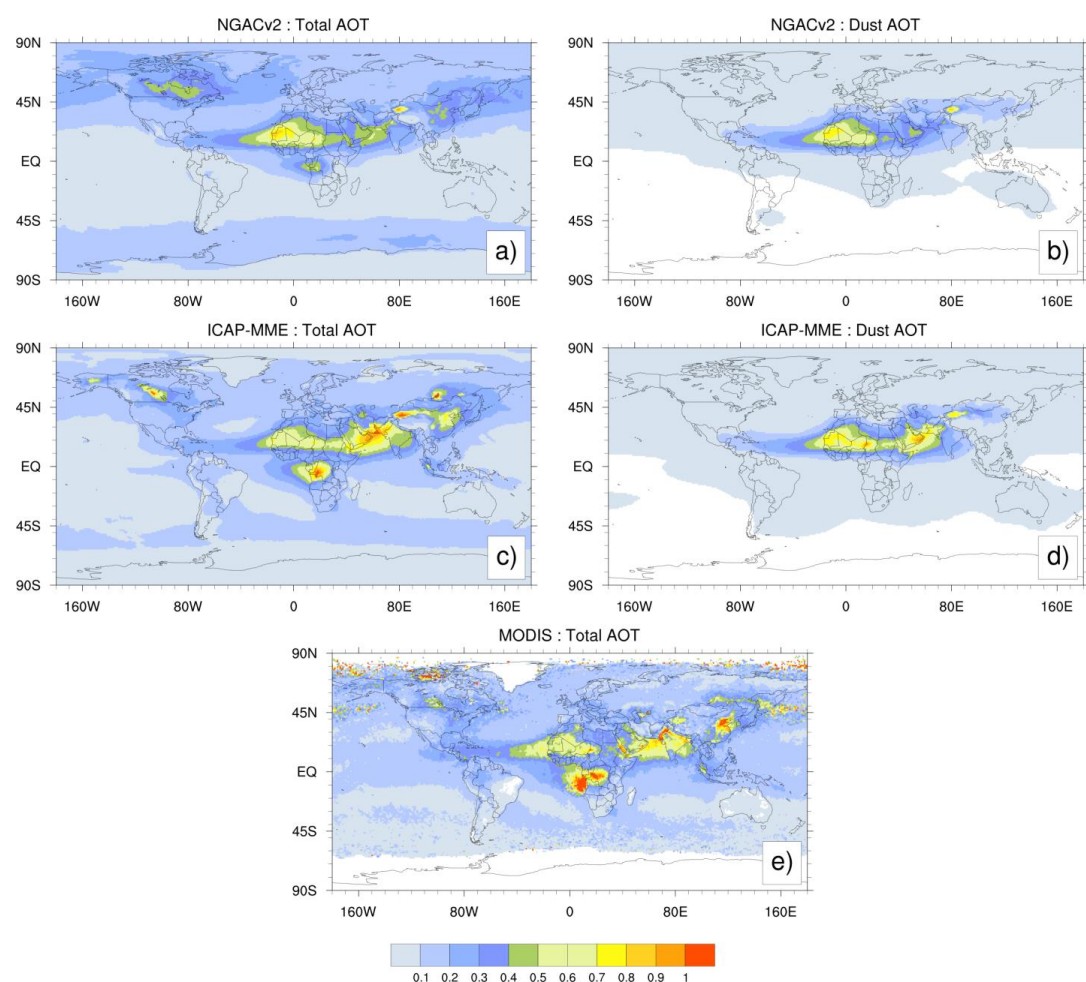

**Figure 1. Global maps of averaged AOT 550nm for JJA (June-July-August) 2015. Total AOT from NGACv2, ICAP and MODIS are in (a, c, e), NGACv2 dust-only (b), and ICAP dust-only (d). NGAC, ICAP and MODIS AOT 550nm are at 1º resolution. Values beyond the range of the color bar are represented by the end colors.**



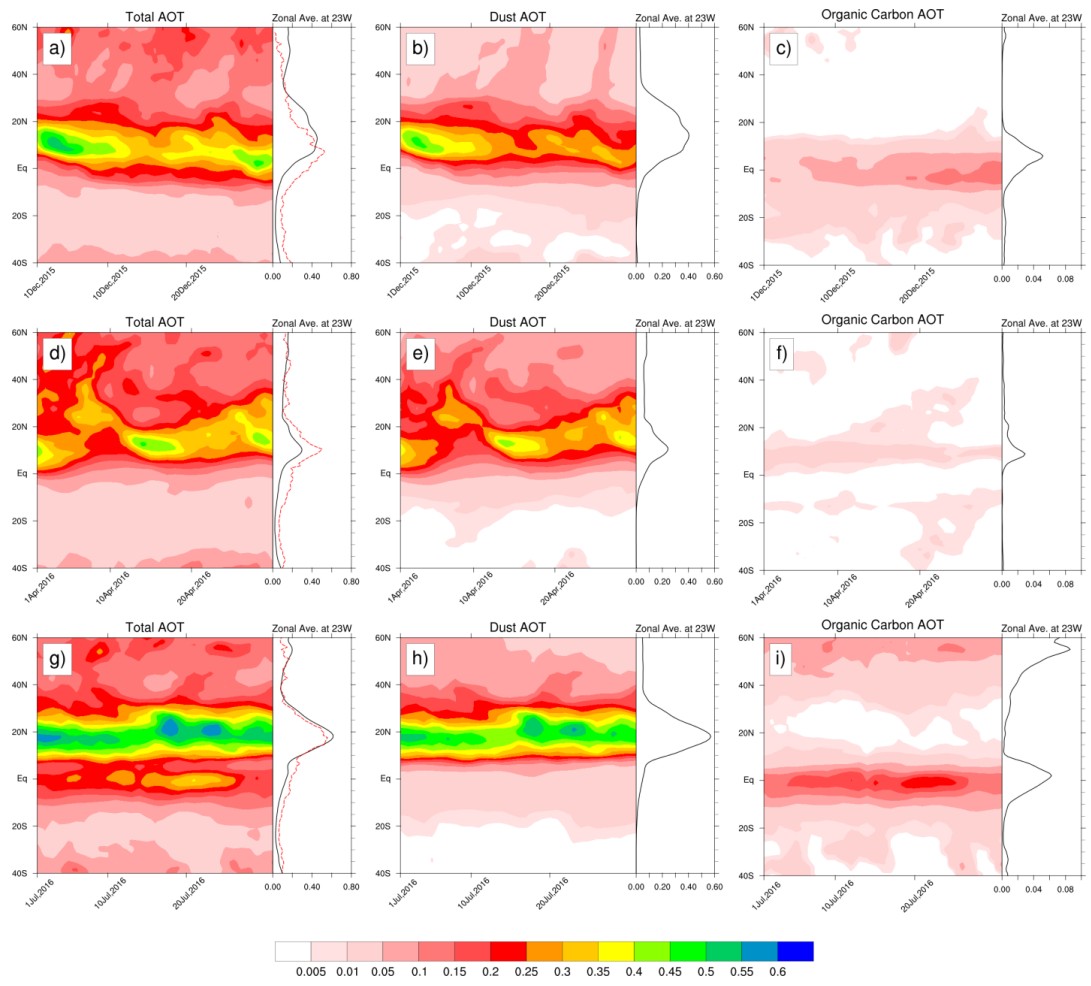

**Figure 2. Latitude-time Hovmöller plot of NGACv2 Total, dust and OC AOT (all at 550nm) over the Atlantic Ocean, zonally averaged (between 60ºW and 30ºE). Top row (a, b, c) is for December 2015, middle row (d, e, f) for April 2016 and bottom row (g, h, i) for July 2016. Line plots show zonal average of Total, dust and OC AOT at 23ºW (over Atlantic Ocean). NGACv2 (black) and MODIS total AOT (red) line in the line plots.**





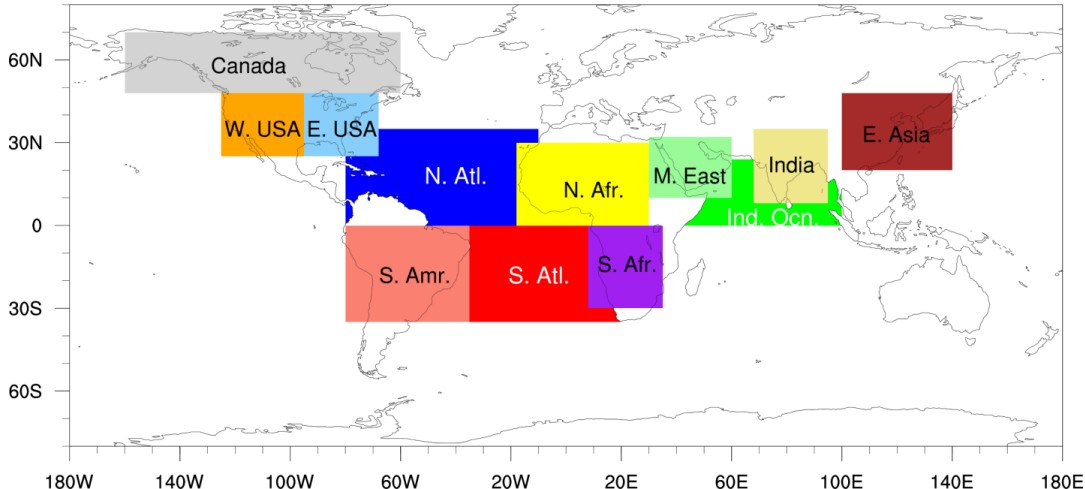

**Figure 3. Map of twelve global zones selected for aerosol analysis between NGACv2, MODIS and ICAP. Details about each zone are described in Table 1.**



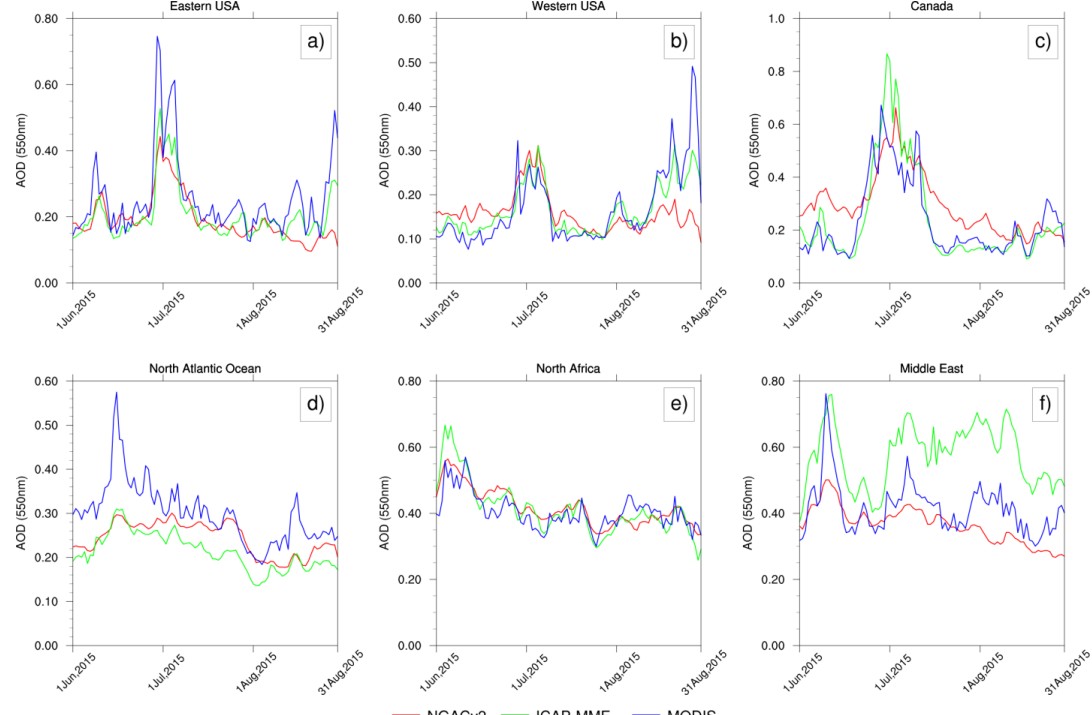

**Figure 4. Regional time-series comparison of daily AOT of modeled and satellite retrieved between June 1st and August 31st, 2015 over selected regions (a-f). See Table 1 for description of the regions. Points over the ocean are masked for calculating AOT over land only regions, and vice versa.**





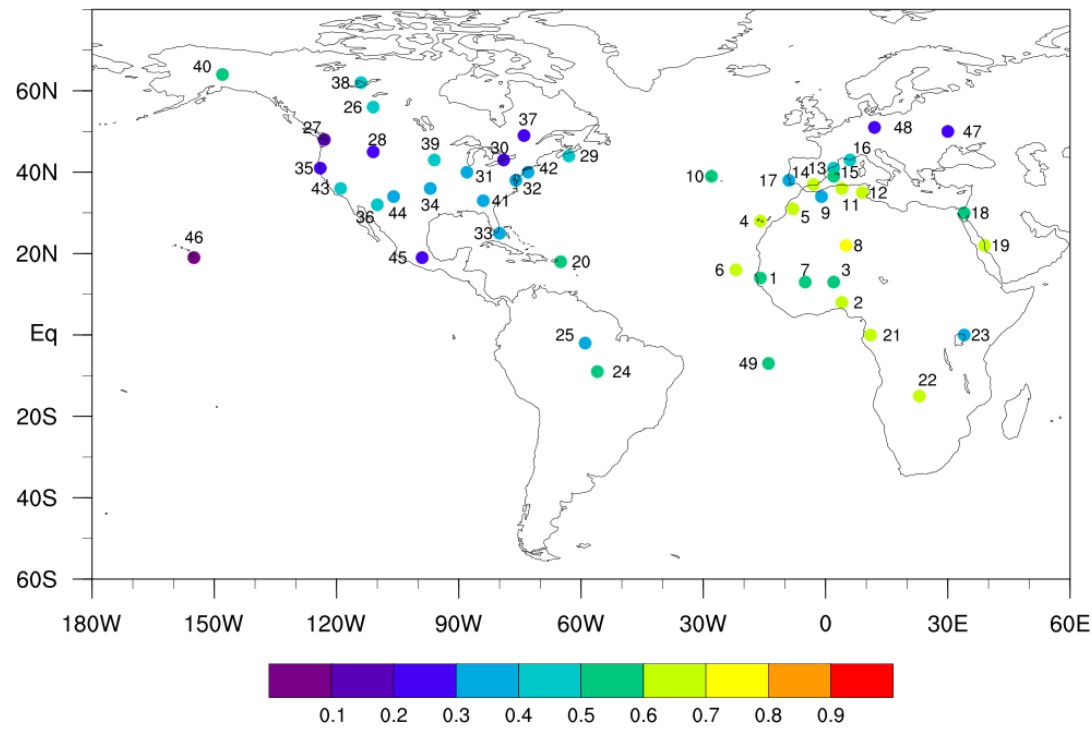

**Figure 5**. **Correlation map of total AOT at 550nm between NGACv2 and AERONET sites. Approximate location of AEORNET centers in the map represented as filled circles. Name and location of these sites are listed in Table 2.**

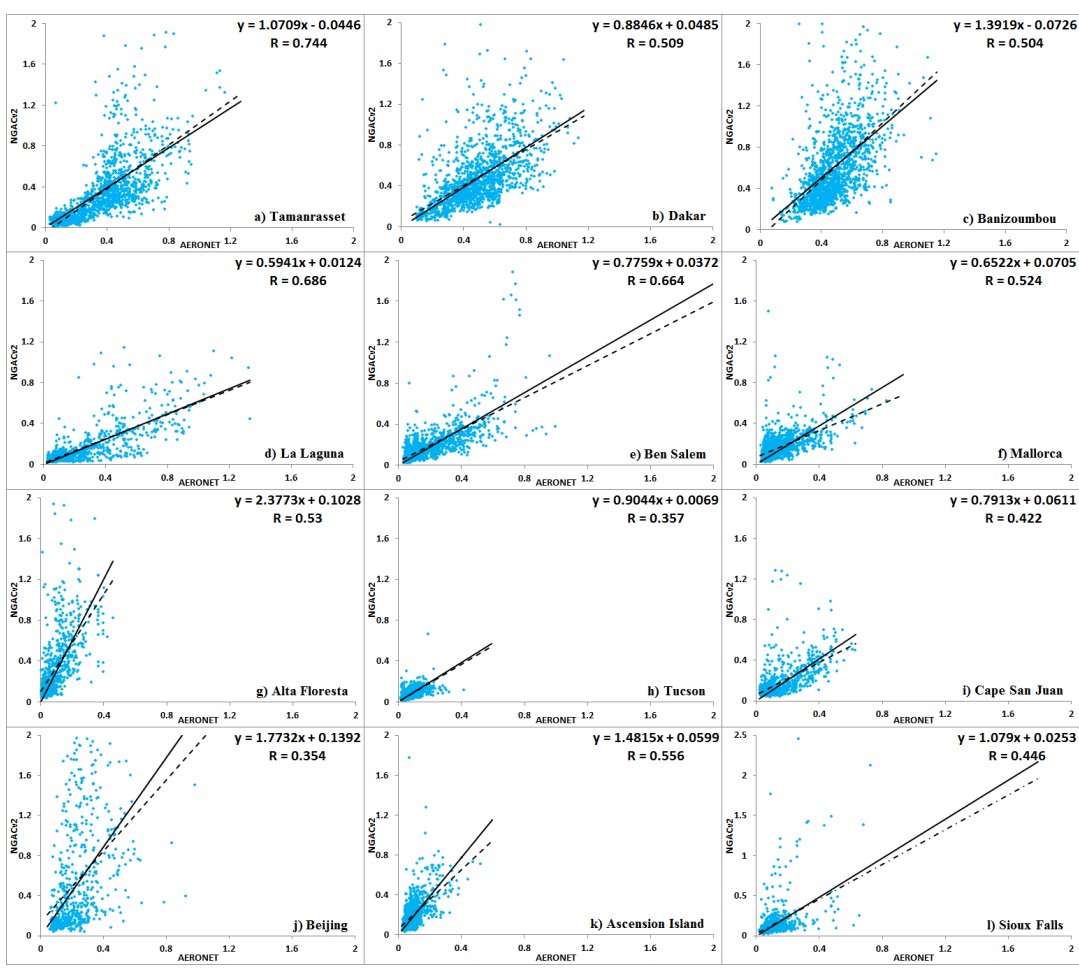

**Figure 6. Correlation plots of 550nm AOT between NGACv2 and 12 AEORNET locations. Black continuous lines in the figures represent the 1:1 line, while dotted black lines represent linear regression fits to data points. Actual locations of AERONET centers are listed in Table 2.**





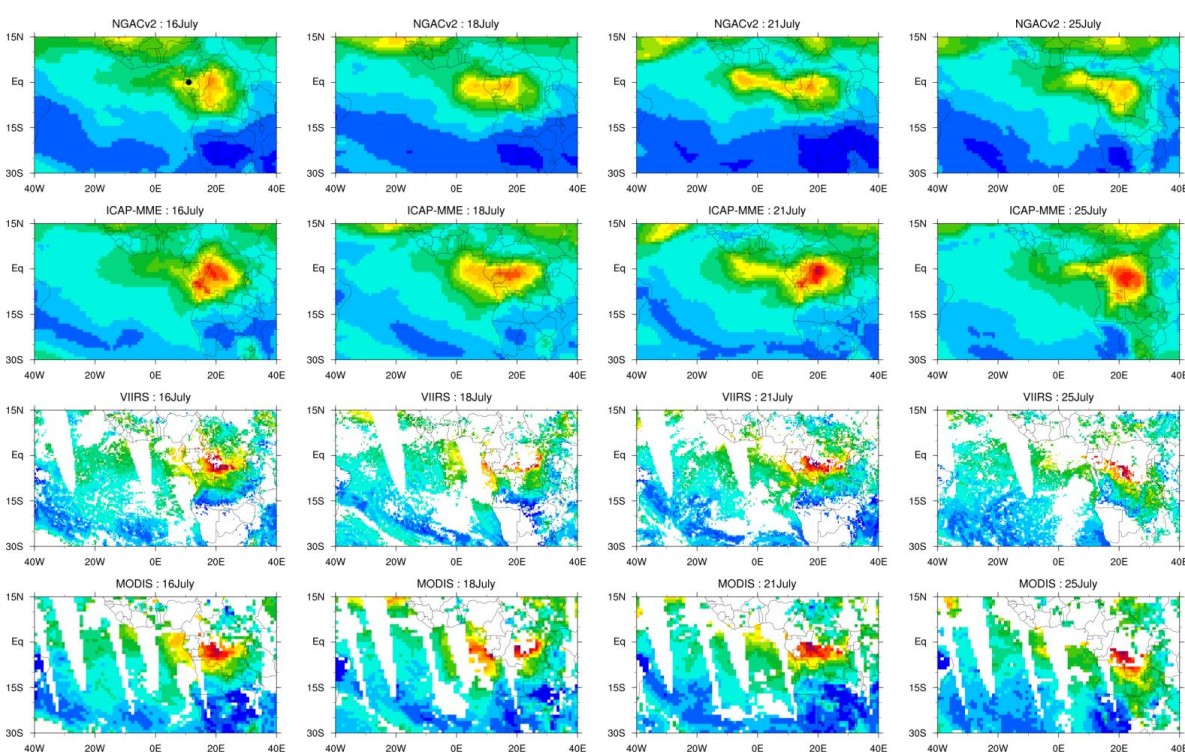

**Figure 7. Comparison of total AOD between forecasts of NGACv2 and ICAP-MME against observations of VIIRS and MODIS for selected days in July 2016. For both models, daily 6-hourly forecasts are averaged to compare against daily satellite observations for each day. Apart from VIIRS, which is at 0.25 degree resolution, all others are at 1 degree. Satellite observations have data gaps, which are in white. Black dot in the first figure represents the approximate location of AERONET station Gabon.**





**Figure 8. Comparison of 550nm AOD between NGACv2 and AERONET location at Gabon for the month of July 2016.**
**Blue line represents total AOD, green line is OC and brown line represents BC AOD, all from NGACv2. Red asterisk**
**symbol is for AERONET observations at that location. AERONET station location is marked in Figure 7.**



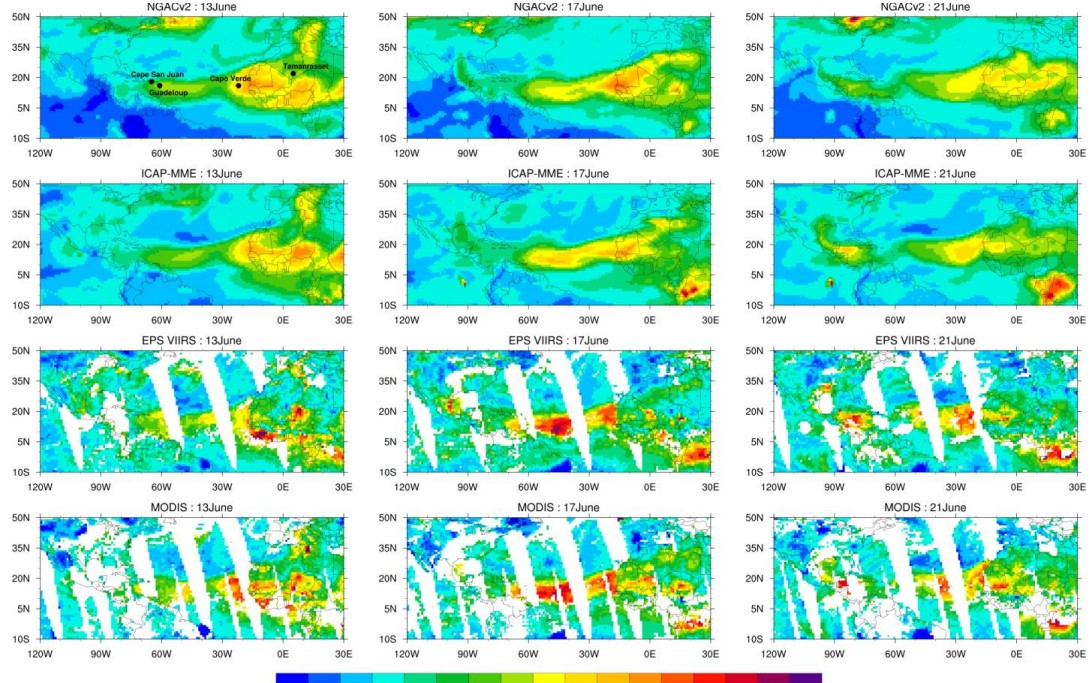

**Figure 9. Comparison of total AOD between forecasts of NGACv2 and ICAP-MME against observations of VIIRS and MODIS for selected days in June 2015. Satellite observations have data gaps, which are in white. Black dots in the first figure represent approximate locations of AERONET stations.**

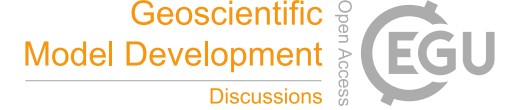



**Figure 10. Comparison of 550nm AOT between NGACv2 and four AERONET locations for the month of June 2015. Blue line represents total AOT, green line is for dust and brown line is for OC AOT, all from NGACv2. Red asterisk symbol is for AERONET observations at that location. AERONET station locations are marked in Figure 9a.**

