# Peer review of "The implementation of NEMS GFS Aerosol Component (NGAC) Version 2.0 for global multispecies forecasting at NOAA/NCEP: Part II Evaluation of Aerosol Optical Thickness"

_Geoscientific Model Development, 2017_

## Referee Comment (RC1) · Anonymous Referee #1 · 19 Jan 2018

General comments

This paper presents the evaluation of the performance of NGACv2, the upgraded NEMS GFS Aerosol Component. The evaluations are mostly performed with observed AOT comparison with multi-model product. The description of the model is very simple because it is written in the companion paper (gmd-2017-306). The evaluation methods are simple and conventional RMSE and correlation factor against observations.

The model uses a well-established GOCART, which makes it reliable as a operational

forecast model but not very innovative. The evaluation methods are conventional but not new approaches. If there are some original characteristics or new concepts in the model, the authors should highlight that part.

Specific comments

p.5, line 19: "AERONET AOT at 440 nm and 675 nm were linearly interpolated on a log-log scale to provide 550nm AOT": Isn't the 500 nm wavelength used for the estimation of AOT at 550 nm?

Section 5: What are the standards or criteria for the scores, especially for correlation coefficient? For example, generally R = 0.28 seems low but the authors descript it is moderate (p.9, line 34).

p.12, line 13: "The model underestimates AOT over the Amazon region in both years and also for the Indonesian fire event in 2015.": It is better to include a presentation of the Indonesian fire event in 2015 in the result or case study section.

Technical corrections

p.1, line 15: 3-dimensioanl -> 3-dimensional

p.2, line 10-11: I don't think the paper of Tanaka and Chiba (2005) is about data assimilation (Probably confused with Sekiyama et al. (2010,ACP)?).

p.3 line 6: "... but is is also ...": there is one extra "is".

p.3 line 13: "... observations.": there is strange bar over the period.

p.3, line 34 and p.4 line 9: "Wang et al., 2017" should be "Wang et al. (2017)"

p.4, line 35 and after: References are separated by commas, the others are separated by semicolons.

p.5, line 3: MASSINGAR -> MASINGAR

p.5, line 6: UKMO's model is just written "(UKMO)" while other models are written with

names as "(Institution-ModelName)".

p.6, line 31: "the s kind that ...": the "s" may be unnecessary?

p.11, line 6 and after: Please check whether "Capo Verde" is correct, or typo of "Cape Verde".

p.4, line 28 and p.13 line 10: SNPP or S-NPP?
* * *

---

## Referee Comment (RC2) · Anonymous Referee #2 · 7 Feb 2018

This is a companion paper to Part I, which presents a new version of NGAC including a fuller range of aerosol species compared to the previous dust-only version. In this paper, a thorough evaluation of the new model is presented, demonstrating good performance in many cases but also identifying a number of errors and biases that might be addressed in a subsequent version. I would recommend publication in GMD subject to the following major comment and further minor comments below

**Major comment.** I would draw particular attention to the Section 5 discussing Figure

[Figure]

6, where a number of aspects of the seasonal cycle are stated in the text, while the figure doesn't appear to present any seasonal information. I would strongly recommend to ensure that the necessary information to visualise these seasonal effects is included in the figures.

**p.1, line 23:** data assimilation does not *improve* the actual model bias, it merely compensates for it in the overall forecasting system (which would still be expected to perform better if the model were unbiased).

**p.2, line 22:** the MACC project has now transitioned into the operational Copernicus Atmosphere Monitoring Service (CAMS).

**p.3, line 32:** I'm not sure "represents an efficient way of transitioning research. . . " is particularly relevant for a model evaluation paper.

**p.4, line 8:** I would suggest "as well as" rather than "apart from" – the latter suggests that mixing ratios are the exception that is *not* available.

**p.4, line 18:** as mentioned later in the paper, using gridded Level-3 data may result in sampling errors compared to matching the model output with Level-2 data; given the known limitations it would be good to explain the reasons for and implications of this choice.

**p.5, lines 1–2:** as above, MACC is now CAMS.

**p.5, lines 10–12:** if "no horizontal interpolation was needed", this is only because it has already been performed upstream – not because the data is being used at its native resolution.

**p.5, lines 23–24:** please explain the rationale for the choice of 2.5 as a threshold, and quantify the proportion of data excluded.

**p.6, line 1:** It is suggested that Figure 1 shows seasonal variation to be in qualitative agreement, but this figure only appears to show one season so what is the evidence for this statement?

**p.6, line 5:** The text says South Africa (usually referring to the specific country), but from the plot it looks more like southern Africa in general is probably meant? Please check and clarify if necessary.

**p.6, lines 8–10:** Given these sampling errors, why not work why not collocate the model output with level-2 data?

**p.6, line 13:** why are only dust and OC shown in Figure 2? Is this because other species can be considered negligible here?

**p.6, line 31:** "of the s kind" – please check!

**p.7, lines 10–11:** agrees in general but the peak is overly broad.

**p.7, line 20:** also very low correlation in MAM16 as well as high RMSE.

**p.8, lines 9–13:** it's unclear why increased cloud thickness would lead to reduced regions of high humidity and thus less hygroscopic growth and lower AOT.

**p.8, lines 22–23:** where is it shown that the model overestimates AOT during November–March? I can't see this in the figure referred to here (Fig. 6).

**p.9, lines 9–10:** again, where is it shown that the model underestimates AOT in September–November?

**p.9, lines 15–16:** again, where is it shown that the model closely reproduces this higher AOT?

**p.9, line 24:** Should this be "The remaining 13 sites" rather than "the rest of the 13 sites" (which would suggest there are only 13 in total)?

**p.9, lines 25–26:** again, where is it shown that modelled AOT is higher than AERONET in May–October?

**p.9, lines 35–36:** where is this underestimation during the summer months shown?

**p.10, line 14:** Figure 8 is not properly introduced until the following paragraph. Also, it might be worth including the ICAP MME in this one.

**p.10, line 23:** "some. . . are higher than the model" – this is over-optimistic; from the figure it appears that almost all are, and some significantly.

**p.11, line 12:** Quantify "correlates well" with an $r$ value.

**p.11, lines 15–16:** the peaks coincide, but the matching of intensity is quite variable.

**p.11, line 24:** it is not obvious why combining multiple observation sources increase uncertainty; much retrieval and data assimilation theory is about using these to *reduce* the resulting uncertainty.

**p.11, line 29:** ICAP-MME is not observations, but an ensemble of models.

**p.12, line 14:** Should be "a" large underestimation, not "at".

**p.13, line 4:** a link or reference should be provided for NCL.

**Table 1:** This table is quite hard to digest; consider presenting visually e.g. with a Taylor diagram.
* * *

---

## Author Comment (AC1) · 17 Mar 2018

Response to the Reviewer 1 comments. Original comments are in bold italics, our response is in regular font.

*General comments*

***This paper presents the evaluation of the performance of NGACv2, the upgraded NEMS GFS Aerosol Component. The evaluations are mostly performed with observed AOT comparison with multi-model product. The description of the model is very simple because it is written in the companion paper (gmd-2017-306). The evaluation methods are simple and conventional RMSE and correlation factor against observations. The model uses a well-established GOCART, which makes it reliable as an operational forecast model but not very innovative. The evaluation methods are conventional but not new approaches. If there are some original characteristics or new concepts in the model, the authors should highlight that part***.

We greatly appreciate the Reviewer's positive comments. We believe addressing the comments listed below resulted in a significantly improved presentation. The Reviewer's effort is greatly appreciated.

*Specific comments*

***p.5, line 19: "AERONET AOT at 440 nm and 675 nm were linearly interpolated on a loglog scale to provide 550nm AOT": Isn't the 500 nm wavelength used for the estimation of AOT at 550 nm?***

We agree with reviewer comment about 500nm wavelength which is used as a standard to compare against other models and satellite observations. However, we found not all AERONET stations that are used in this study report 500nm wavelength AOT values at regular basis (Level 1.5 data). To make all comparisons consistent, we used linear interpolation between 440 and 675 nm (which are reported in all stations) to derive 550 nm AERONET AOT. Also, MODIS and ICAP-MME that are used in this study provide AOT at 550nm.

Revised manuscript: Page 5, line 18 added this sentence "NGACv2 outputs AOT at 550nm and several AEORNET sites do not report at 500 or 550nm wavelengths".

*Section 5: What are the standards or criteria for the scores, especially for correlation coefficient? For example, generally R = 0.28 seems low but the authors descript it is moderate (p.9, line 34).*

We have used number of sample points and location of AEORNET center for correlation coefficient criteria. In this case, number of sample points is low (523 from Table 2) for 17 month comparison study. Also, location of this AERONET center is at remote Southern Indian Ocean and no other AERONET centers available near it to validate the observation. Based on these two criteria, we have described the correlation as "moderate" in this case.

Revised manuscript : Page 5, line 27, added this sentence "We have given more weightage on number of sample points and AERONET location (Table 2) for qualitatively describing correlation coefficients at each location as "low", "moderate" and "high" in this study".

*p.12, line 13: "The model underestimates AOT over the Amazon region in both years and also for the Indonesian fire event in 2015.": It is better to include a presentation of the Indonesian fire event in 2015 in the result or case study section.*

We evaluated our model against observations and other models for massive Indonesia fire of 2015. Major cause of NGACv2 underestimation is due to very low emission in GBBEPx detected in this region that has been used in the model. Currently we are conducting experiments

with different emissions and Data assimilation to improve model result over this fire event. Above is one figure that show improvement in AOD forecast over Indonesia (left one without any DA, right hand one is with DA). The one with DA, represent Indonesian fire better than the one without DA.

***Technical corrections***

***p.1, line 15: 3-dimensioanl -> 3-dimensional***

Corrected spelling in the manuscript.

Revised manuscript: at page 1, line 15.

***p.2, line 10-11: I don't think the paper of Tanaka and Chiba (2005) is about data assimilation (Probably confused with Sekiyama et al. (2010,ACP)?).***

We corrected and removed reference from Tanaka and Chiba (2005) from that line and reference list. We have reference of Sekiyama et al 2010 in that line and reference list.

Revised manuscript: page 2, line 10 added reference of Sekiyama et al 2010. Also Page 17, Line 12 added the reference of Sekiyama et al 2010.

*p.3 line 6: "... but is is also ...": there is one extra "is".*

Corrected.

Revised manuscript: at page 2, line 6.

*p.3 line 13: "... observations.": there is strange bar over the period.*

Corrected.

Revised manuscript: at page 3, line 13.

*p.3, line 34 and p.4 line 9: "Wang et al., 2017" should be "Wang et al. (2017)"*

Corrected in both pages.

Revised manuscript: at page 3 lines 15 and 34.

*p.4, line 35 and after: References are separated by commas, the others are separated by semicolons.*

Corrected.

Revised manuscript: at  page 4 line 35.

*p.5, line 3: MASSINGAR -> MASINGAR*

Corrected.

Revised manuscript: at page 5, line 3.

*p.5, line 6: UKMO's model is just written "(UKMO)" while other models are written with names as "(Institution-ModelName)".*

Corrected.

Revised manuscript: at page 5, line 6.

*p.6, line 31: "the s kind that ...": the "s" may be unnecessary?*

Corrected.

Revised manuscript: at  Page 6, line 34.

*p.11, line 6 and after: Please check whether "Capo Verde" is correct, or typo of "Cape Verde".*

We used AERONET station name (which is Capo Verde) in the manuscript, it can be found here at AERONET official site: https://aeronet.gsfc.nasa.gov/aeronet_locations_v3.txt.

*p.4, line 28 and p.13 line 10: SNPP or S-NPP?*

Corrected to S-NPP in both pages.

Revised manuscript: at page 4 line 28 and page 13 line 25.

---

## Author Comment (AC2) · 17 Mar 2018

Response to the Reviewer 2 comments. Original comments are in bold italics, our response is in regular font.

***General comments***

***This is a companion paper to Part I, which presents a new version of NGAC including a fuller range of aerosol species compared to the previous dust-only version. In this paper, a thorough evaluation of the new model is presented, demonstrating good performance in many cases but also identifying a number of errors and biases that might be addressed in a subsequent version. I would recommend publication in GMD subject to the following major comment and further minor comments below***.

We greatly appreciate Reviewer's comments. We believe addressing the comments listed below resulted in a significantly improved presentation. The Reviewer's effort is greatly appreciated,

***Specific comments***

***Major comment. I would draw particular attention to the Section 5 discussing Figure 6, where a number of aspects of the seasonal cycle are stated in the text, while the figure doesn't appear to present any seasonal information. I would strongly recommend to ensure that the necessary information to visualise these seasonal effects is included in the figures***.

We thank the reviewer for pointing this out and we added a new figure in the manuscript to address this comment. New figure 8 in the manuscript shows time-series of NGACv2 and AERONET stations (correspond to same stations in new Figure number 7). Time-series plot show seasonal variation of both dataset that is mentioned in the text and answer to reviewer's other questions related to now figure 7 (in the questions below). We also have corrected manuscript text (in Section 5) and added few lines/sentences where reference to Figure 8 is mentioned with new figure numbers. Subsequent changes in the other figure numbers in the manuscript are made.

Revised manuscript : Page 8, line 35 added this sentence "Figure 8 shows entire 17 month time-series of AOT at same twelve stations shown in Figure 7 between the two". Also added Figure 8 in page 29.

*p.1, line 23: data assimilation does not improve the actual model bias, it merely compensates for it in the overall forecasting system (which would still be expected to perform better if the model were unbiased).*

We used the term "bias" to broadly include any type of error that is systematic rather than random. We followed reviewer suggestion and edited the sentence by removing "improve some of the model biases" from that sentence and added "provide positive impact in the aerosol forecast by the model".

Revised manuscript: Page 1, line 23, added "provide positive impact in the aerosol forecast by the model".

*p.2, line 22: the MACC project has now transitioned into the operational Copernicus Atmosphere Monitoring Service (CAMS).*

We made the correction in the text.

Revised manuscript: at page 2, line 22.

*p.3, line 32: I'm not sure "represents an efficient way of transitioning research. . . " is particularly relevant for a model evaluation paper.*

We described implementation of new global forecast model at NOAA, which is joint collaboration with NASA is an example of Research to Operations (R2O).

*p.4, line 8: I would suggest "as well as" rather than "apart from" – the latter suggests that mixing ratios are the exception that is not available.*

Corrected.

Revised manuscript: at page 4, line 8.

*p.4, line 18: as mentioned later in the paper, using gridded Level-3 data may result in sampling errors compared to matching the model output with Level-2 data; given the known limitations it would be good to explain the reasons for and implications of this choice*.

We agree with reviewer comment that using Level-2 satellite swaths coincident with model forecast time would have produced less sampling error. However, in this paper our main focus was to check long-term model forecast in global regions and we looked into major aerosol events in general. With subsequent upgrade of the model we will conduct more detailed comparison with other dataset using coincident observation samples in near future.

*p.5, lines 1–2: as above, MACC is now CAMS*.

Corrected.

Revised manuscript: at page 5, line 2.

*p.5, lines 10–12: if "no horizontal interpolation was needed", this is only because it has already been performed upstream – not because the data is being used at its native resolution*.

We agree with reviewer comment.

Revised manuscript: page 5, line 11, we have removed "no horizontal interpolation was needed to put the different data sources onto a single grid" from the sentence.

*p.5, lines 23–24: please explain the rationale for the choice of 2.5 as a threshold, and quantify the proportion of data excluded*.

We found some of AERONET station data during high aerosol events report AOT values (range from 3-8), which would be impossible by the model to correctly forecast (due to resolution). We used the threshold of 2.5 which correspond to lower number of data to be excluded for statistical calculation. Nearly 3% of sample points are excluded with this threshold.

Revised manuscript: at page 5, line 24-26, we have added "Some of the station data report AOT of 5 and above in extreme high aerosol events (smoke and pollution transport) which may not be simulated by the model due to coarse resolution. We estimated approximately 3% of the data are discarded due to this threshold".

*p.6, line 1: It is suggested that Figure 1 shows seasonal variation to be in qualitative agreement, but this figure only appears to show one season so what is the evidence for this statement?*

We looked into all the seasons between 2015 and 2016 for comparison with ICAP and MODIS. We selected one representative season (2015 JJA) to show seasonal variation in the figure 1. Correlation and RMSE for all six seasons are described in Table 1. Our statement of seasonal variation based on both Figure 1 and Table 1 statistics.

Revised manuscript: at Page 6, line 3, we have added "We analyzed model results for both 2015 and 2016 and Figure 1 shows results from 2015".

*p.6, line 5: The text says South Africa (usually referring to the specific country), but from the plot it looks more like southern Africa in general is probably meant? Please check and clarify if necessary*.

We thank reviewer for pointing this and we corrected the sentence with "Southern Africa". We selected biomass burning zone south of equatorial Africa for Figure 1 and table 1.

Revised manuscript: Page 6 line 12, corrected South Africa with "southern Africa".

*p.6, lines 8–10: Given these sampling errors, why not work why not collocate the model output with level-2 data?*

As mentioned before, we agree with reviewer statement. However, collocation of observation in Level 2 data and matching that with model forecast time is beyond the scope of this study. We focus on general model characteristic over different aerosol regimes with new capabilities of detecting biomass burning aerosols. Also, present simulations cover six seasons that correspond to the time period when GBBEPx emissions are generated for pre-implementation evaluation. The collocated approach for this relatively short period of time could lead to a small sampling size due to the collation screening.

*p.6, line 13: why are only dust and OC shown in Figure 2? Is this because other species can be considered negligible here?*

Dust and smoke aerosols shown in Figure 2 are dominant aerosols for the region considered. We picked OC as a representative of biomass burning aerosol in the figure to show model sensitivity in this region and compared with observation.

*p.6, line 31: "of the s kind" – please check!*

Corrected.

Revised manuscript: Page 7, line 2.

*p.7, lines 10–11: agrees in general but the peak is overly broad*.

We agree with reviewer comment on this.

Revised manuscript: at page 7, line 19-20 we added this sentence "However, NGACv2 simulated peak is broad compare to other two indicate model is less sensitive to capture some of AOT variations over Canada (Figure 4c)".

*p.7, line 20: also very low correlation in MAM16 as well as high RMSE*.

We agree with reviewer comment on low correlation and high RMSE about middle-east dust AOT in 2016. Dust intensity in 2015 and 2016 are different and model seems to capture 2015 event better than 2016.

*p.8, lines 9–13: it's unclear why increased cloud thickness would lead to reduced regions of high humidity and thus less hygroscopic growth and lower AOT*.

We have made correction in the text and removed the sentence "The GFS also tends to overestimate cloud layer thickness, particularly for deep convective clouds in the tropical regions" from the text to avoid any confusion. In this paragraph we discussed possible reasons for model's underestimation of AOT over South-East Asia during summer months. Study by Yoo et al. (2013) which is referred in that section diagnose GFS biases in simulating low-cloud fractions against satellite and ground observations.

Revised manuscript: Page 8, line 27 we have removed this sentence "The GFS also tends to overestimate cloud layer thickness, particularly for deep convective clouds in the tropical regions".

*p.8, lines 22–23: where is it shown that the model overestimates AOT during November– March? I can't see this in the figure referred to here (Fig. 6).*

We added new figure (8) in the manuscript that show model overestimation over AERONET location Tamanrasset between November and March (Figure 8a).

Revised manuscript : Page 29, added Figure 8 that correspond to all locations described in Figure 7. Also added reference to figure in page 9, line 5.

*p.9, lines 9–10: again, where is it shown that the model underestimates AOT in September– November?*

We added new figure (8) that show model underestimation over AERONET location Alta Floresta (in South America). The underestimation is during the forest fire season over Brazil (Figure 8g).

Revised manuscript : Page 29, added Figure 8 that correspond to all locations described in Figure 7. Also added reference to figure in page 9, line 27.

*p.9, lines 15–16: again, where is it shown that the model closely reproduces this higher AOT?*

We added time-series figure of Fort McMurray (located in North America) for the entire 17month period here. It compares NGACv2 daily forecast (blue line) and AERONET observation (red points) at that location. The model captures high biomass burning events during summer of 2015 and 2016 and peak intensity matches with AERONET observation and correspond to description in the manuscript.

[Figure]

Fort Mcmurray (56N,111W)

*p.9, line 24: Should this be "The remaining 13 sites" rather than "the rest of the 13 sites" (which would suggest there are only 13 in total)?*

Corrected.

Revised manuscript : Page 10, line 7 corrected the sentence.

*p.9, lines 25–26: again, where is it shown that modelled AOT is higher than AERONET in May–October?*

We added time-series figure of Mauna Lua (located in Hawai) for the entire 17month period here. It compares NGACv2 daily forecast (blue line) and AERONET observation (red points) at that location. It shows the model overestimate AOT for the entire time-period against ground observations and correspond to description in the manuscript.

[Figure]

***p.9, lines 35–36: where is this underestimation during the summer months shown?***

We added time-series figure of Mexico City for the entire 17month period. It compares NGACv2 daily forecast (blue line) and AERONET observation (red points) at that location. It shows the model underestimates AOT for majority of the time period and underestimation is higher during summer months and correspond to the description in the manuscript.

[Figure]

*p.10, line 14: Figure 8 is not properly introduced until the following paragraph. Also, it might be worth including the ICAP MME in this one*.

We included 6-hourly forecast of ICAP-MME in Figure 10 (revised figure number) over the same AEORNET location shown in Figure 9. Gaps in ICAP-MME forecast is due to use of 6 hourly data, whereas NGACv2 is at every 3 hour.

Revised manuscript : Added ICAP-MME at Page 32 Figure 10. Added reference to figure in page 11, line 1.

*p.10, line 23: "some. . . are higher than the model" – this is over-optimistic; from the figure it appears that almost all are, and some significantly*.

NGACv2 underestimate initial biomass burning episode between $1^{st}$ and $14^{th}$ of July, 2016 (Figure 9). Model's peak intensity is lower than AERONET and ICAP-MME during that time. However, as shown in Figure 9, when more intense burning reported (between $16^{th}$ to $25^{th}$ July) and observed over Southern Africa , model AOT intensity matches closely with ICAP-MME which is at same 1º resolution (Figure 10).

*p.11, line 12: Quantify "correlates well" with an r value*.

We added correlation (r) value of that location in the manuscript.

Revised manuscript : at page 11, line 33.

*p.11, lines 15–16: the peaks coincide, but the matching of intensity is quite variable*.

We agree with reviewer comment about intensity. This difference may also arise from mismatch between model grid compared against a point location.

Revised manuscript: Page 12, line 1-2 we added "model simulated intensity is lower compare to the observation".

*p.11, line 24: it is not obvious why combining multiple observation sources increase uncertainty; much retrieval and data assimilation theory is about using these to reduce the resulting uncertainty*.

We stated about multiple observations in the context of comparing against model forecast. As different observations contain different calibrations, resolution, wavelengths for AOT product to consider before comparison against the model. In this preliminary study with new capabilities of the model, our primary focus is to look into any systematic error by the model in the study period. That is why we have used only MODIS satellite data for AOT and avoided any other uncertainty associated with comparing against different satellite products. We would like to use multiple satellite data in future for more advanced study with model AOT forecast.

***p.11, line 29: ICAP-MME is not observations, but an ensemble of models***.

We mentioned ICAP-MME as multi-model ensemble in that sentence and different than satellite or other observations.

Revised manuscript : at page 12, line 9.

***p.12, line 14: Should be "a" large underestimation, not "at".***

Corrected.

Revised manuscript : Page 12 line 35.

***p.13, line 4: a link or reference should be provided for NCL***.

We added a link to NCL home page in the manuscript.

Revised manuscript : at Page 13, line 26.

***Table 1: This table is quite hard to digest; consider presenting visually e.g. with a Taylor diagram***

We thank reviewer for the suggestion. We have added Taylor diagram (Figure number 5) to show seasonal variations at same locations mentioned in Table 1.

Revised manuscript : at Page 26 we added Figure 5. Also, added description about the figure at page 8, line 8-16 as "We used Taylor diagrams to summarize model performance in different seasons over the same regions described in Table 1 (Figure 5). Taylor diagrams (Taylor, 2001) provide a statistical summary of comparisons between NGACv2 and MODIS observations in

terms of their spatial correlation coefficients and the ratio of spatial standard deviations of the model and observations over all twelve regions. The spatial correlation coefficient is the quantity that measures the degree of agreement of two fields and standard deviations are normalized by the corresponding observations. In general, model's performance is better in summer months (JJA, Figure 5a) than other seasons in terms of low variance and high correlations over most of the regions. However, in future we needed a more detailed study to understand some of the interannual variations shown by the model, particularly over land regions (Figure 5)".

---

## Author Response (AR2)

Response to the Reviewer comments. Original comments are in bold italics, our response is in regular font.

We greatly appreciate Reviewer's positive effort in overall improvement of the manuscript.

***Specific Suggestion***

***In addition, I'd like to make a remark on the name "Capo Verde". You are right that this is indeed how AERONET spells the station name. Nevertheless, the name is not correct. It should be either "Cape Verde" (English) or "Cabo Verde" (Portuguese). I suggest to use the correct name of the country in the text. When referring to the AERONET station, you could use "Capo Verde" in parentheses***

We corrected the manuscript and changed all references of Capo Verde to Cape Verde as per reviewer suggestion. Only in Table 2, where AERONET center names are listed, we have used "Capo Verde".

***Specific Comments***

***The authors have appropriately addressed most of the discussion comments, and this revised version is a significant improvement, in particular with the new Figures 5 and 8 (per-season Taylor diagrams and station time-series plots) now allowing the seasonal effects described in the text to be seen more clearly.***

***However, I do have a few outstanding comments that should be addressed before final publication in ACP:***

***Regarding reviewer 1's comment on Section 5 and the description of R=0.28 as moderate, the revision doesn't really address the main points of either statistical significance (is this a dataset where R=0.28 could occur by chance alone?) or fraction of variance explained. Similarly, at page 11, line 33, r=0.375 is now described as "correlates well". These both seem like low r values compared to many of those in Tables 1 and 2, so the criteria for what should be considered a good correlation need to be clearer.***

We determined the statistical significance of correlation between model and AEORNET observation in these two instances and found correlations are not significant at 95% confidence intervals.

Revised manuscript: Page 10, line 16-17, we rephrased the paragraph as "Sea-salt aerosol is dominant over remote Amsterdam Island in the southern Indian Ocean and model correlation is low (R=0.28) at 95% confidence intervals but associated with low RMSE".

Page 12, line 32-34, we rephrased the sentence as "At Cape Verde, which is located just off the coast of Africa, NGACv2 correlation is low (R=0.375) at 95% confidence interval with AERONET observations, and also overestimates the intensity (nearly 2 times) during the event (Figure 12b)".

***Reviewer 1's comment about including the 2015 Indonesian fire event in the case studies, given that it's referred to in the conclusions, doesn't seem to have been acted upon. The manuscript would certainly be improved by including this, along with the explanation given in the author's response that the underestimation results from the emission dataset.***

We have added a new figure (Figure 13) and a description about Indonesia forest fire in the manuscript.

Revised manuscript : Page 13, line 3-14 we have added "The 2015 fire season in Indonesia started in July and lasted through October with haze extended through Malaysia, Singapore, and Thailand and exposed millions of people to hazardously poor air quality (Field et al. 2016). Figure 13 shows total AOT from NGACv2, ICAP-MME and MERRA2 forecasts compared against EPS-VIIRS observation on a single day in September, 2015 over south-east Asia. 6-hourly model forecasts are averaged to get daily AOT for the models. NGACv2 underestimates total AOT which is caused by low smoke emission (both OC and BC) data used by the model for this fire event. Wei et al (2017) analyzed both forecast and analysis of MERRA2 aerosol fields and compared that against NGACv2. That study also compared aerosol analysis increments (defined as difference between analysis and model first guess) of all four cycles of MERRA2 and found large AOT analysis increment (0.6-0.8) in 06z DA cycle which contributed to higher AOT in MERRA2. Thus, the underestimation of Indonesian fire by NGACv2 can be attributed to both

near-real time emissions and absence of DA. Lynch et al (2016) study showed that AOT DA is as equally important as tuning process of the sources and sinks of aerosols."

We added Figure 13 in Page 35 of the manuscript. Also, we have added new references in the reference list.

*At page 8, lines 8–9, it would be good to mention the good agreement with the ICAP-MME as supporting the theory that the significant underestimation for this case is a generic feature of coarse-resolution models, as suggested in the response to reviewer 2's comment. "Some" here should still be replaced with "many" or similar. Against AERONET (rather than ICAP-MME) it is the majority of data points in Figure 10 that are too low.*

We corrected the text and replaced "some" with "majority" in that sentence.

*And a couple of minor technical corrections:*

*Page 5, lines 1–2: although the acronym "MACC" has been updated to "CAMS" as per reviewer 2's comment, the full version has not. It should be "European Centre for Medium-Range Weather Forecasts / Copernicus Atmosphere Monitoring Service".*

Correction made at the manuscript.

Revised manuscript : Page 5, line 1-2 corrected to "the European Centre Medium Range Weather Forecasts Copernicus Atmosphere Monitoring Service (ECMWF-CAMS)".

*Page 5, line 18: the addition contains a typo (AEORNET should be AERONET).*

We have corrected this particular spelling in the text, figure captions.